# Obesity in Saudi Medical Students and Alignment with Eating Disorders: A Stratified Random Cross-Sectional Investigation

**DOI:** 10.3390/healthcare12131281

**Published:** 2024-06-27

**Authors:** Abdullah A. Alrasheed, Nasser M. AbuDujain, Samar Almohammedi, Rahaf Alrayes, Fahad Alajlan, Osama Abdulqader, Majd Albarrak, Khalid H. Alharbi, Turky H. Almigbal, Mohammed A. Batais

**Affiliations:** 1Department of Family and Community Medicine, College of Medicine, King Saud University, Riyadh 11495, Saudi Arabia; aalrasheed1@ksu.edu.sa (A.A.A.); khalharbi@ksu.edu.sa (K.H.A.); talmigbal@ksu.edu.sa (T.H.A.); mbatais@ksu.edu.sa (M.A.B.); 2College of Medicine, King Saud University, Riyadh 11495, Saudi Arabia

**Keywords:** medical students, obesity, eating disorders, SCOFF, Saudi Arabia

## Abstract

As a result of the increasing global incidence of obesity and related diseases, this study aims to investigate the prevalence of obesity and its correlation with eating disorder (ED) screening among medical students to develop effective prevention strategies and provide better outcomes for these students. We conducted a quantitative analytical cross-sectional study at the College of Medicine at King Saud University between November 2023 and February 2024. A stratified random sampling technique was utilized, enrolling a net number of 415 participants. Participants were asked several questions, including demographic information, weight, height, and past medical history, and were given a validated screening tool for EDs. Participants’ mean age was 21.4 ± 1.67 years, and 17.6% were considered obese. The prevalence of positive screening for EDs was 27.7%; it was more common in females (*p* = 0.013) and those who earned more than 1000 SAR per month (*p* = 0.011). Female students also exhibited almost twice the odds of having EDs than males (AOR = 1.957; 95% CI = 1.218–3.146; *p* = 0.006). Furthermore, non-obese students showed decreased odds of having EDs compared with obese students by at least 48% (AOR = 0.517; 95% CI = 0.287–0.929; *p* = 0.027). Our study revealed a concerning prevalence of ED symptoms and obesity among medical students, suggesting profound implications. Therefore, multicenter studies are needed to assess the generalizability of the results and apply the findings to targeted national campaigns and interventions tailored specifically to medical students.

## 1. Introduction

Eating disorders (EDs) are fairly serious to life-threatening conditions defined as maladaptive eating practices that majorly impact physical and psychological health [1]. Anorexia nervosa (AN), bulimia nervosa (BN), and binge-eating disorder (BED) are identified by the American Psychiatric Association (APA) as the most common global EDs [1]. Such disorders are prevalent among young people, according to the World Health Organization (WHO). In 2019, 14 million people were affected by EDs, including almost 3 million children and adolescents [2].

Young adults between the ages of 18 and 25 are vulnerable to societal pressures, including peer influence, societal expectations regarding appearance, and the impact of social media, all of which can lead to adopting detrimental eating habits [3]. A cross-sectional study conducted in the U.S. between 2013 and 2021 aimed to determine whether ED risk has increased among U.S. college students. It found that the risk of eating disorders among over 260,000 U.S. undergraduate students increased significantly from 15% in 2013 to 28% in 2020/2021 [4].

Locally, a face-to-face national survey was conducted in Saudi Arabia in 2022, with over 4000 participants ages 15–65, to measure the prevalence of AN, BN, and BED in Saudi citizens. The results revealed that the 12-month prevalence of any of these three EDs was 3.2%, with an overall lifetime prevalence of 6.1%; this is higher than rates reported globally [5]. This study also found a significant association between the body mass index (BMI) and the lifetime prevalence of BED and BN. Such research presents a connection between EDs and obesity, which is defined as an excessive accumulation of body fat above the normal range, as indicated by a BMI of 30 or higher for adults [6]. It is linked to a variety of diseases and poses significant health risks, including type-2 diabetes mellitus, non-alcoholic fatty liver disease, and cardiovascular disease [7,8]. A systematic review investigates the prevalence of eating disorders and disordered eating behaviors (EDs/DEB) among Saudi Arabian adolescents and youth, analyzing findings from 14 papers from five databases. Prevalence rates of EDs/DEB in 11 studies varied between 10.2% and 48.1%, with the eastern region showing the highest rates ranging from 29.4% to 65.5%. The study reveals that older students have a higher prevalence than younger ones, and males demonstrate a greater incidence of EDs/DEB compared to females [9].

The literature also shows an international escalating trend in the prevalence of obesity. According to the most recent data from WHO, in 2022, 16% of adults over 18 were obese; this is more than double the number of obese people worldwide in 1990 [6]. Furthermore, a correlation between Saudi Arabian prevalence with WHO from the years of 2000 to 2022 shows a concerning rise, as revealed in Figure 1. A recent review in Saudi Arabia showed a similar pattern on the rise, with a current 35.6% obesity rate compared with the 1990–1992 prevalence of 22% [10,11]. Additionally, a systematic review done in Saudi Arabia reported a 30% prevalence of obesity among young adults [12]. This increase is concerning, and studies have shown similar patterns among highly health-educated individuals, namely medical students. Several international investigations to assess the prevalence of obesity in medical students revealed a range of 5.2–31.67% [13,14,15]. Similarly, a cross-sectional study conducted among healthcare science college students revealed that 11% of participants were obese. The prevalence of obesity was higher among males [16].

Moreover, regarding the link between obesity and EDs, a large retrospective cohort study conducted in Spain between 2001 and 2010 aimed to determine the prevalence of obesity in ED subtypes. It found a rate of 5% in cases of AN, 33.2% in cases of BN, and 87% in cases of BED. Furthermore, the prevalence of lifetime obesity among ED patients has increased twofold in the last decade [17].

Similarly, there is a high prevalence of EDs among medical students. A recent large meta-analysis of the prevalence of EDs and their associated risk factors in medical students found a pooled prevalence rate of 17.35%. This is higher than in the general population. The analysis also evaluated the relationship between EDs and obesity in medical students. Its findings supported the hypothesis that elevated BMI is linked to heightened symptoms of EDs, thus highlighting the need for future epidemiological research to explore the relationship between BMI and ED risk. [18].

To help address this gap, our study investigates the prevalence of obesity and its correlation with EDs among medical students to develop effective prevention strategies and provide better outcomes for this population, which represents an important pillar of future health care. By focusing on this crucial area of study, we aspire to provide valuable insights into the prevalence and risk factors of, as well as potential interventions for, EDs and obesity within this specific group.

## 2. Methodology

### 2.1. Study Design, Participants, and Setting

We conducted a quantitative analytical cross-sectional study between November 2023 and February 2024 at the College of Medicine at King Saud University in Riyadh, Saudi Arabia. The target population consisted of male and female full-time medical students registered in that semester. The college administration provided a list of serial numbers for all regular students for further randomization and recruitment. The total number of students was 1488 (905 male and 583 female).

After calculating the sample size using the known population formula, with a 5% margin of error and 95% confidence interval (CI), it was determined that a minimum of 319 participants was needed for the study. However, to account for potential non-responses, 600 participants were included. To ensure a representative sample, a stratified random sampling method was used. This involved collecting the official university registry number for all medical students and dividing them into subgroups based on sex. Random sampling was then achieved by assigning a random number to each student using Microsoft Excel’s “= RAND()” command. The data were then sorted from smallest to largest, and the first 300 numbers from each subgroup were selected for inclusion.

### 2.2. Instruments

The survey created for this study consisted of two parts. The first collected demographic and personal history data, while the second comprised the SCOFF questionnaire, a pre-validated patient-reported outcome tool commonly used to screen for EDs.

#### 2.2.1. The Sociodemographic Questionnaire 

This section consisted of two parts. The first one gathered demographic information such as sex, age, nationality, academic level, and income. The second part obtained details of participants’ personal history, such as height, weight, and any previous diagnoses of chronic illnesses. Obesity was screened with the known BMI formula (weight in kilograms divided by the square of the person’s height in meters [kg/m^2^]), which was later calculated during data entry.

#### 2.2.2. SCOFF Questionnaire

The SCOFF questionnaire is a simple five-question tool developed by Morgan et al. in 2002 and is widely used to assess EDs [19]. It consists of five brief yes/no questions addressing important features of EDs, such as vomiting, feeling fat, losing control over one’s eating habits, weight loss, and whether food takes precedence in one’s life; every “yes” answer counts for 1 point and every “no” for 0. A total score of 2 or more indicates the possibility of an individual having AN or BN. The Arabic version of SCOFF (A-SCOFF), validated by Aoun et al., has a sensitivity of 80% and a specificity of 72%. Permission to use it was obtained from the authors [20].

### 2.3. Procedure and Data Collection 

The Institutional Review Board approved this study at the College of Medicine at King Saud University as Project No. E-23-8055 (Ref. No. 23/0679/IRB). An online survey was created through the SurveyMonkey website, and a link was sent to all participants. The nature and purpose of the study, the primary investigator’s contact information, an explanation of the confidentiality and data anonymity policy, and the Institutional Review Board’s approval number were provided. Consent to participate was given by clicking on the informed consent link. After reading the informed consent statement, participants clicked on “next” to access the study’s survey, which took approximately 5–7 min to complete.

### 2.4. Statistical Data Analysis

Categorical variables were described as counts and proportions (%), while continuous variables were calculated and expressed as means and standard deviations (SDs). The relationship between obesity and EDs, according to the sociodemographics of medical students, was conducted using the Chi-square test. Significant results were then tested in multivariate regression analyses to determine the independent significant risk factors for obesity and EDs. A *p*-value of less than 0.05 was considered statistically significant. All statistical data were analyzed using Statistical Packages for Social Sciences version 26 (IBM Corp., Armonk, NY, USA).

## 3. Results

A total of 415 medical students were enrolled in the study. Their mean age was 21.4 (SD 1.67), with 52% between 18 and 21. About half were female (50.1%), and most were Saudi (98.6%). The average monthly income was 1643.3 SAR (438 USD), and nearly half (49.9%) were earning more than 1000 SAR (267 USD). Third-year students constituted 22.9%. In addition, 17.6% were considered obese, with 79.5% of them being male and 20.5% being female, while 17.3% had a previous history of medical conditions, with depression (24.2%) and anxiety (21%) being the most common (Table 1).

Regarding the assessment of the SCOFF questionnaire (Table 2), the highest ratings were given for losing control of eating (37.6%), the belief that food dominates life (24.8%), and the belief that oneself is fat although others would say the opposite (21.4%). The overall mean SCOFF score was 0.92 (SD 1.08). Accordingly, 27.7% were considered positive for EDs, and the rest were negative (72.3%).

Measuring the relationship between EDs and the sociodemographic characteristics of medical students (Table 3) revealed that the prevalence of ED symptoms was statistically significantly more common among females (*p* = 0.013), those who earned more than 1000 SAR monthly (*p* = 0.011), those who were obese (*p* = 0.025), and students with a previous history of medical conditions (*p* = 0.020). When evaluating the relationship between BMI levels and sociodemographic characteristics of the medical students (Table 4), it was found that those who were obese were more likely to be older (*p* = 0.010), male (*p* < 0.001), and senior students (*p* = 0.028).

As presented in regression model 1, compared with male students, females exhibited at least 1.96 times higher odds of having EDs (AOR = 1.957; 95% CI = 1.218–3.146; *p* = 0.006), and students earning more than 1000 SAR per month had at least 1.73-fold higher odds of having EDs (AOR = 1.732; 95% CI = 1.109–2.707; *p* = 0.016). However, compared with obese students, non-obese students showed decreased odds of having EDs by at least 48% (AOR = 0.517; 95% CI = 0.287–0.929; *p* = 0.027). As shown in regression model 2, compared with males, female students were at a decreased risk of at least 79% of being obese (AOR = 0.211; 95% CI = 0.115–0.388; *p* < 0.001). Finally, age and academic year had no significant effect on BMI after adjustment to a regression model (*p* > 0.05; Table 5).

## 4. Discussion

This study aimed to determine the prevalence and risk factors of ED symptoms and identify the association between obesity and EDs among medical students at King Saud University. The findings showed a high prevalence of obesity, affecting 17.6% of the students. Moreover, the SCOFF questionnaire revealed that 27.7% screened positive for EDs. Notably, the analysis revealed relationships among gender, EDs, and weight, with female students exhibiting higher odds of EDs, while their male counterparts faced increased risks of obesity. Additionally, obese students and those with higher incomes seemed more susceptible to EDs.

The study also revealed a strikingly high prevalence of EDs among medical students, with 27.7% screening positive based on the SCOFF questionnaire. This rate aligns with findings from similar studies, such as one conducted at King Abdulaziz University in Jeddah, Saudi Arabia, where 32.1% of medical students screened positive for EDs [21]. Additionally, in Dammam, Saudi Arabia, 29.4% of preparatory-year college students were categorized as having a high level of EDs [22], and at the College of Medicine in Taif, Saudi Arabia, a substantial 27.9% of medical students were identified as being at high risk for EDs [23]. While these studies employed the Eating Attitudes Test (Eat-26) for screening, the research results align with existing evidence, indicating a positive correlation between SCOFF and Eat-26 [9].

Internationally, though with slightly lower percentages, the prevalence of EDs also remains concerning. For instance, 17% of medical students in Karachi, Pakistan, screened positive for them using the SCOFF questionnaire [24]. Likewise, in a Lebanese medical school, 19% of medical students exhibited positive screenings for EDs, and in Puerto Rico, 9.59% of college students screened positive [25,26]. This high prevalence of ED symptoms can be attributed to high stress levels among medical students, who face increased levels of anxiety compared with the general global population [27]. Numerous factors contribute to the escalating distress experienced as medical training advances. Among these are the intensely competitive learning atmosphere, the systems for grading and evaluation, and an implicit curriculum that fosters unhealthy comparisons with peers, thereby reinforcing perfectionistic tendencies. These factors, in turn, elevate students’ susceptibility to developing EDs [28,29].

The 17.6% prevalence of obesity among medical students found in this study was interconnected with a complex relationship between obesity and EDs, as highlighted by parallels in prior research. For instance, Ghamri et al.’s study [21] supports these findings, underscoring the propensity for overweight and obese individuals to exhibit high-risk dietary behaviors, aligning with this study’s observed tendencies among medical students.

Additionally, similar results were seen in Alwosaifer et al.’s and Ali et al.’s studies in Saudi Arabia [22,30], as well as across diverse geographies, as evidenced by studies in Karachi, Pakistan, and several U.S. universities [24,31], where the prevalence of ED screenings was notably elevated among overweight and obese people. Furthermore, a similar trend was discerned among medical students at Damascus University [32]. However, a contrasting narrative emerged from a study in Iran [33], challenging the assumption that BMI serves as a reliable predictor of EDs among medical sciences students.

The presumed correlation between EDs and BMI is challenged when considering the complex psychological underpinnings of such conditions, which extend beyond mere body weight. Nonetheless, this association is clarified when examining the profound impact of body weight on body image. The literature has linked obesity to heightened vulnerability to negative body image, a significant risk factor for developing EDs [34]. Furthermore, obese individuals are at a higher risk of experiencing low self-esteem, which in turn elevates their risk of having such disorders [35].

In addition, female medical students exhibited a heightened susceptibility to EDs compared with their male counterparts, a trend echoed by local studies conducted in Jeddah and Dammam, Saudi Arabia [21]. This gender disparity finds support in Yaqoob et al.’s research in China, which underscores the correlation between the female sex and EDs. However, Alrahili et al.’s study in Abha presented a contrasting perspective, noting the absence of a clear correlation between being female and the risk of EDs [36]. Similarly, Tsekoura et al.’s investigation in Greece yielded inconclusive results [37]. However, the above studies focused on children and adolescents, potentially accounting for the discrepancies observed.

The intricate interplay between gender and EDs finds an explanation in societal pressures and psychological factors. Females often contend with heightened familial and peer expectations surrounding idealized body image, fostering fertile ground for the development of disordered eating behaviors [38]. Moreover, the elevated risk of anxiety and depression among females serves as a well-established precursor to EDs [39], further exacerbating vulnerability.

Finally, this study revealed a positive association between higher income levels and ED risk among medical students. While, to our knowledge, no prior research in Saudi Arabia has explored the relationship between socioeconomic status and EDs, existing literature suggests that college students from affluent backgrounds are more inclined to perceive the need for ED treatment [40].

The findings of this study have several implications for medical education and student well-being. First, the high prevalence of EDs and obesity among medical students underscores the urgent need for mental health and nutritional support services tailored to this population. Moreover, incorporating topics related to mental health, body image, and nutrition into the medical curriculum can enhance awareness and equip medical students with the skills to deal with these issues and ask for support if needed. Additionally, community engagement, including family education about the pressure faced by medical students and the importance of giving them social support, is essential.

### Limitations

This study has several limitations that warrant consideration. First, its focus solely on medical students at one college in a single urban center restricts the generalizability of the findings to a broader population. Second, the reliance on self-reported data introduced potential sources of bias. Similarly, calculating BMI based on self-reported height and weight introduces another layer of potential error; however, a point that may overcome this is that the studied sample consists of medical students, which increases the accuracy of the entered variables due to their routine health checks. Lastly, the genetic heredity of obesity was not evaluated. For future research studying this topic, we recommend undertaking an in-person interview which includes measurement of height and weight by one of the research investigators, as well as a full assessment for eating disorders using wider tools for diagnostic purposes.

## 5. Conclusions

This study revealed a concerning prevalence of ED screenings and obesity among medical students, echoing findings in existing literature and highlighting a significant correlation between these two conditions. The implications are profound, calling for targeted national campaigns and interventions tailored specifically to medical students. Prioritizing positive body image, fostering healthy relationships with food and exercise, and ensuring accessible support resources are pivotal steps in navigating the intricate interplay between obesity and EDs among this demographic. Furthermore, engendering collaboration among universities, healthcare providers, and community organizations is paramount in creating nurturing environments that prioritize student well-being. Universities should implement policies that promote student well-being, including stress management programs, healthy lifestyle workshops, and accessible mental health resources.

To aid in these measures, future research endeavors should aim for broader representation across universities and cities, more deeply analyzing the underlying drivers of these conditions and exploring longitudinal trajectories to inform more comprehensive interventions and support systems.

## Figures and Tables

**Figure 1 healthcare-12-01281-f001:**
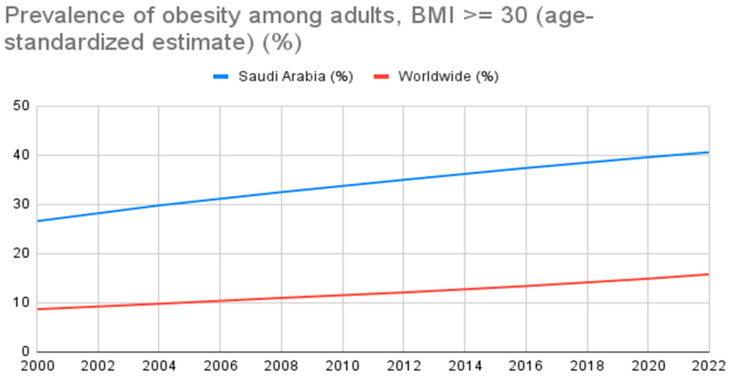
The trend of obesity in Saudi Arabia from 2000 until 2022, compared with the worldwide results reported by the World Health Organization.

**Table 1 healthcare-12-01281-t001:** Sociodemographic characteristics of participants ^(n = 415)^.

Sociodemographic	N (%)
Age group (mean ± SD)	21.4 ± 1.67
18–21 years	216 (52.0%)
>21 years	199 (48.0%)
Gender	
Male	207 (49.9%)
Female	208 (50.1%)
Monthly income in SAR (mean ± SD)	1643.3 ± 1057.3
≤1000 (267 USD)	208 (50.1%)
>1000 (267 USD)	207 (49.9%)
Academic year	
1st Year	74 (17.8%)
2nd Year	83 (20.0%)
3rd Year	95 (22.9%)
4th Year	74 (17.8%)
5th Year	89 (21.4%)
BMI level	
Underweight (<18.49 kg/m^2^)	44 (10.6%)
Normal (18.5–24.9 kg/m^2^)	203 (48.9%)
Overweight (25–29.9 kg/m^2^)	95 (22.9%)
Obese (≥30 kg/m^2^)	73 (17.6%)
**Medical history**	
Previous history of medical condition	
Yes	72 (17.3%)
No	343 (82.7%)
Chronic illnesses diagnosed	
Depression	15 (24.2%)
Generalized anxiety disorder	13 (21.0%)
Gastroesophageal reflux disease	07 (11.3%)
Dyslipidemia	04 (06.5%)
Hypothyroidism	04 (06.5%)
Type 1 Diabetes Mellitus	03 (04.8%)
Polycystic ovarian syndrome	03 (04.8%)
Irritable bowel syndrome	03 (04.8%)
Asthma	02 (03.2%)
Other *	08 (12.8%)

* Include hypertension, cholelithiasis, rheumatoid arthritis, bipolar disorder, epilepsy, G6PD deficiency, tetralogy of Fallot, and migraine.

**Table 2 healthcare-12-01281-t002:** Descriptive statistics of SCOFF questionnaire ^(n = 415)^.

Variables	N (%)
1. Do you make yourself sick because you feel uncomfortably full?	31 (07.5%)
2. Do you worry you have lost control over how much you eat?	156 (37.6%)
3. Have you recently lost more than one stone in a 3-month period?	04 (01.0%)
4. Do you believe yourself to be fat when others say you are too thin?	89 (21.4%)
5. Would you say that food dominates your life?	103 (24.8%)
Eating disorder	
Positive (score ≥ 2)	115 (27.7%)
Negative (score < 2)	300 (72.3%)

**Table 3 healthcare-12-01281-t003:** Relationship between eating disorders and the sociodemographic characteristics of medical students ^(n = 415)^.

Factor	Eating Disorder		*p*-Value ^§^
	Positive N (%) (n = 115)	Negative N (%) (n = 300)	
Age group			
18–21 years	58 (50.4%)	158 (52.7%)	0.684
>21 years	57 (49.6%)	142 (47.3%)	
Gender			
Male	46 (40.0%)	161 (53.7%)	**0.013 ***
Female	69 (60.0%)	139 (46.3%)	
Monthly income (SAR)			
≤1000 (267 USD)	46 (40.0%)	162 (54.0%)	**0.011 ***
>1000 (267 USD)	69 (60.0%)	138 (46.0%)	
Academic year			
Junior students (1st–3rd year)	69 (60.0%)	183 (61.0%)	0.852
Senior students (4th–5th year)	46 (40.0%)	117 (39.0%)	
BMI level			
Obese	28 (24.3%)	45 (15.0%)	**0.025 ***
Non-obese	87 (75.7%)	255 (85.0%)	
Previous history of medical condition			
Yes	28 (24.3%)	44 (14.7%)	**0.020 ***
No	87 (75.7%)	256 (85.3%)	
Previous diagnosis of mood disorder			
Yes	11 (09.6%)	19 (06.3%)	0.255
No	104 (90.4%)	281 (93.7%)	

^§^ *p*-value has been calculated using Chi-square-test. * Significant at *p* < 0.05 level.

**Table 4 healthcare-12-01281-t004:** Relationship between obesity and the sociodemographic characteristics of medical students ^(n = 415)^.

Factor	Level of BMI		*p*-Value ^§^
	Obese N (%) (n = 73)	Non-Obese N (%) (n = 342)	
Age group			
18–21 years	28 (38.4%)	188 (55.0%)	**0.010 ***
>21 years	45 (61.6%)	154 (45.0%)	
Gender			
Male	58 (79.5%)	149 (43.6%)	**<0.001 ***
Female	15 (20.5%)	193 (56.4%)	
Monthly income (SAR)			
≤1000 (267 USD)	37 (50.7%)	171 (50.0%)	0.915
>1000 (267 USD)	36 (49.3%)	171 (50.0%)	
Academic year			
Junior students (1st–3rd year)	36 (49.3%)	216 (63.2%)	**0.028 ***
Senior students (4th–5th year)	37 (50.7%)	126 (36.8%)	
Previous history of medical condition			
Yes	11 (15.1%)	61 (17.8%)	0.571
No	62 (84.9%)	281 (82.2%)	
Previous diagnosis of mental disorder			
Yes	02 (02.7%)	28 (08.2%)	0.103
No	71 (97.3%)	314 (91.8%)	

^§^ *p*-value has been calculated using Chi-square-test. * Significant at *p* < 0.05 level.

**Table 5 healthcare-12-01281-t005:** Multivariate regression analysis to determine the significant independent risk factors associated with obesity and eating disorders ^(n = 415)^.

Model 1: Eating Disorders	AOR	95% CI	*p*-Value
Gender			
Male	Ref		
Female	1.957	1.218–3.146	**0.006**
Monthly income (SAR)			
≤1000 (267 USD)	Ref		
>1000 (267 USD)	1.732	1.109–2.707	**0.016 ***
BMI level			
Obese	Ref		
Non-obese	0.517	0.287–0.929	**0.027 ***
Previous history of medical condition			
Yes	1.611	0.930–2.789	0.089
No	Ref		
Model 2: Obesity			
Age group			
18–21 years	Ref		
>21 years	1.544	0.685–3.480	0.294
Gender			
Male	Ref		
Female	0.211	0.115–0.388	**<0.001 ***
Academic year			
Junior students (1st–3rd year)	Ref		
Senior students (4th–5th year)	1.127	0.506–2.508	0.770

AOR: Adjusted Odds Ratio; CI: Confidence Interval. * Significant at *p* < 0.05 level.

## Data Availability

The data used during the current study are available from the corresponding author upon reasonable request.

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
