# Peer review of "Obesity in Saudi Medical Students and Alignment with Eating Disorders: A Stratified Random Cross-Sectional Investigation"

_healthcare, 2024, doi:10.3390/healthcare12131281_

Round 1

Reviewer 1 Report

Comments and Suggestions for Authors

Study is interesting and quite relevant. Some suggestions and recommendations are provided.

1. would suggests to provide some sort of a chart or graph representing obesity (in comparison with other countries or timeline - historical perspective) for better clarity - within the introduction; similar with the issue of eating disorder

2. just curious, why would medical students be more cautious and knowledgeable (or health conscious) - stress related with the occupation? or study process?

3. the data analysis is straight forward, statistical treatment seems fine - did you try hierachial linear regression? try to use some control variables? 

was hoping to find more inputs with regards to life style or stress or anxiety levels that might influence and get into why specifically medical students.

findings - what now? provide more practical implications and recommendations

Author Response

Dear Editor,

Thank you for the comprehensive review of our submission and consideration of our paper. We have thoroughly viewed all reviewers’ comments, and it was track-changed in yellow. Furthermore, as requested, the following is an itemized response to all reviewers’ comments.

** Reviewer 1: Thank you for your high-yield comments. As recommended, we have added a chart representing obesity in comparison to the timeline in the introduction. Regarding obesity in medical students, we have elaborated further in the discussions. Regarding analysis, we used the method entered in logistics regression, not the hierarchical linear regression, as linear regression is not appropriate for the data. Moreover, we have now expanded the implication section (last paragraph in the discussion) to provide practical future recommendations.

Reviewer 2 Report

Comments and Suggestions for Authors

Title: Obesity in Saudi Medical Students and Alignment with Eating Disorders: A Stratified Random Cross-Sectional Investigation

 The authors present an investigation into the prevalence of obesity and its correlation with detected eating disorders (ED) among medical students at King Saud University between November 2023 and February 2024.

 While the topic of the study is interesting, the methodology carried out leaves many doubts about the validity of the study, which is why I do not recommend its publication.

 Other comments:

 Line 38. Could the authors narrow the statistic on the prevalence of eating disorders to focus on the adolescent and university-age population studied, and not just that of the 15 - 65 year age group in general?

 Linea 52. Do the authors consider obesity in the population to be an educational problem? Or that knowing more about nutrition prevents obesity? Please provide some argument that would give more weight to your study.

 Also, please add some arguments to the effect that the medical student population is more likely to develop eating disorders and obesity.

 Line 84. How do the authors claim to represent the student population by semesters or year? Different semesters or years better represent the problems of eating disorders related to adaptation issues (first semesters) and exam pressures as students advance through the year. Please add something about this, as this only shows the number of individuals per year to the results of the study and should be indicated in the methodology section.

 Line 98. How is the weight and height of the participants obtained? Is it self-reported? If so, do the authors consider this to be a valid source of anthropometric measurements? I think this is the main limitation of the study.

 Line 102. Do the authors consider the SCOFF questionnaire is a valid diagnostic test? Or is it a screening test for eating disorders?

Author Response

Dear Editor,

Thank you for the comprehensive review of our submission and consideration of our paper. We have thoroughly viewed all reviewers’ comments, and it was track-changed in yellow. Furthermore, as requested, the following is an itemized response to all reviewers’ comments.

** Reviewer 2: Thank you for your valuable comments. As you kindly recommended, we have further discussed young adult age-group statistics and relations in the introduction. Concerning the semester comment, in Saudi Arabia, particularly the location of the study (College of Medicine at King Saud University), medical students and their curriculum do not follow a semester system; rather, blocks (1st and 2nd year), themes (3rd year) or cycles (4th and 5th years); therefore, we have built our classification based of years. However, we agree with the abovementioned comment that the relationship with the exam period could serve as an external stress factor. Therefore, we have mentioned it in the future recommendation section. In regards to nutrient knowledge and obesity, we indeed did not touch on that, as it is not one of our objectives. Concerning the hypothesis of medical students being more likely to have eating disorders, it is further elaborated in the discussion. Regarding the weight and height comment, we asked medical students to manually enter their weight and height. While, as you mentioned, this might hold recall bias, medical students at KSU undergo regular health check appointments (for vaccination and overall health), where their weight and height are measured in the clinic by a staff nurse; we held that they are aware of their last measurements of height and weight. The SCOFF tool is a well-established screening tool for eating disorders, and it is not diagnostic. A positive screening by SCOFF warrants a thorough evaluation by a specialist using other tools like EAT-26; for that, as we have written in the manuscript, the results rely on positive “screening” rather than a positive diagnosis.

We hope that you will review our manuscript and reconsider its suitability for publication.

Reviewer 3 Report

Comments and Suggestions for Authors

In general, it is a well-designed and written research. I have indicated the suggestions that I think are appropriate to make the manuscript more understandable below:

Most of the information presented in the introduction section belongs to the adult age group. However, the target group of the study is young adults who are university students.

It was not stated whether the ethics committee permission was obtained for the study.

It may be more appropriate to write "Chronic illnesses diagnosed" instead of the "Specific medical condition" subtitle in Table 1.

While the prevalence of study variables is presented in Table 1, the proportional distribution of chronic diseases is presented under the heading "Specific medical condition". This leads to the deterioration of the integrity of the table. The prevalence of chronic diseases should be included in this table (e.g. Depression: 3.6%, Polycystic Ovarian Syndrome: 1.4%)

The prevalence of obesity in both sexes should be given separately in the first paragraph of the findings. Although the number of obese students is stated as 73 in Table 1 and Table 4, it is shown as 87+256=343 people in Table 3. Therefore, the relationship between obesity and ED may also have been miscalculated.

Logistic regression is used to calculation of an Odds Ratio which is related to the negative effects of independent variables on the dependent variable. When Odds Ratio smaller than 1, 1/Odds Ratio is used to calculate the risk.

It is expected that there is a positive correlation between age and academic year. In logistic regression, on the other hand, the highly correlated variables should not be included in the same model and only one of these variables should be preferred.

Comments on the Quality of English Language

The manuscript needs revision for language and grammar.

Author Response

Dear Editor,

Thank you for the comprehensive review of our submission and consideration of our paper. We have thoroughly viewed all reviewers’ comments, and it was track-changed in yellow. Furthermore, as requested, the following is an itemized response to all reviewers’ comments.

** Reviewer 3: Thank you for your comments. We have added more information about the target group (young adults) in the introduction. As per the ethical approval, our project has indeed been approved by the institutional review board at King Saud University. We have now added the approval number and information in the methods section. We have rephrased the sentence you recommended into “Chronic illnesses diagnosed”. Regarding separating Table 1 into two separate tables, we originally made them into one table to avoid the abundance of tables. However, we have separated the table into two sections (while still in Table 1) to maintain the integrity of the information. Regarding the prevalence in Table 3, it indeed was an unintentional mistake and has now been rectified. Concerning statistical analysis, we agree regarding the first point. However, both decreased and negative associations are the same in meaning. In the second point, the assumption is written in the statistical method that all significant results were tested on the regression to determine if they are still significant after multivariate regression analysis. Regarding the language editing, we have already submitted the work to a professional language editing service, and they have undergone extensive language editing (we will upload the language editing certificate). Nonetheless, we have sent our manuscript once more to a few bilingual speakers with a good background in syntax, and they provided us with some further enhancements.

Reviewer 4 Report

Comments and Suggestions for Authors

The manuscript 'Obesity in Saudi Medical Students and Alignment with Eating Disorders: A Stratified Random Cross-Sectional Investigation' is a research study that investigates the factors affecting ED in a Saudi medical college. I am happy that the authors covered all the necessary literature to discuss the issue and presented a clear methodology. However, I have only one comment that might limit the study. The authors did not sort out the genetic heredity obesity disorders. It is also helpful if authors can submit a revised version after reducing plagiarism.

Author Response

Dear Editor,

Thank you for the comprehensive review of our submission and consideration of our paper. We have thoroughly viewed all reviewers’ comments, and it was track-changed in yellow. Furthermore, as requested, the following is an itemized response to all reviewers’ comments.

** Reviewer 4: Thank you for your comments. We have mentioned genetic heredity in our limitation section and future research recommendations. Regarding the plagiarism, we have already sent our manuscript to a language editing and reviewed the % plagiarism, which turned out to be < 5% throughout the manuscript; if there are certain sections you would like us to modify, please do let us know.

Round 2

Reviewer 1 Report

Comments and Suggestions for Authors

after going over the revisions made by the author/s, the current version of the paper is clear and is adequate for acceptance

Author Response

Dear Esteemed Reviewer,

Thank you for taking the time to review our manuscript and considering it for publication.

Reviewer 2 Report

Comments and Suggestions for Authors

The authors present a study that aimed to investigate the prevalence of obesity and its correlation with the detection of eating disorders (ED) among medical students at King Saud University between November 2023 and February 2024.

While the topic of the study is interesting, the methodology carried out leaves many doubts about the validity of the study, which is why I do not recommend its publication.

While the title of the article talks about obesity, the data obtained from the participating population was through a questionnaire in which the participants' weight and height were asked. With these data, the Body Mass Index (BMI) was calculated to estimate obesity. This is not valid and takes away all the seriousness of the study, so in my opinion it cannot be accepted for publication.

Author Response

Dear Esteemed Reviewer,

Thank you for re-reviewing our manuscript.

We still firmly believe that the data entered is accurate. As mentioned, medical students studying at King Saud University undergo frequent health assessments, where their height and weight are measured by staff nurses. Nonetheless, we purely understand your concern, and to tackle that, we have added in the future recommendation section for future research studying this topic to undertake an in-person interview, which includes measurement of height and weight, as well as a full assessment for eating disorder using wider tools for diagnostic purposes.

Thank you once again,